# Proteasome inhibition-enhanced fracture repair is associated with increased mesenchymal progenitor cells in mice

Hengwei Zhang[1], Xing Li[1], Jiatong Liu[1], Xi Lin[1], Lingpeng Pei[2], Brendan F. Boyce[1,3], Lianping Xing[1,3]*

1 Department of Pathology and Laboratory Medicine, University of Rochester Medical Center, Rochester, New York, United States of America, 2 Key Laboratory of Ethnomedicine, Minzu University of China, Beijing, China, 3 Center for Musculoskeletal Research, University of Rochester Medical Center, Rochester, New York, United States of America

☯ These authors contributed equally to this work.
* Lianping_xing@urmc.rochester.edu

**Data Availability Statement:** All relevant data are within the manuscript and its Supporting information files.

## Abstract

The ubiquitin/proteasome system controls the stability of Runx2 and JunB, proteins essential for differentiation of mesenchymal progenitor/stem cells (MPCs) to osteoblasts. Local administration of proteasome inhibitor enhances bone fracture healing by accelerating endochondral ossification. However, if a short-term administration of proteasome inhibitor enhances fracture repair and potential mechanisms involved have yet to be exploited. We hypothesize that injury activates the ubiquitin/proteasome system in callus, leading to elevated protein ubiquitination and degradation, decreased MPCs, and impaired fracture healing, which can be prevented by a short-term of proteasome inhibition. We used a tibial fracture model in *Nestin*-GFP reporter mice, in which a subgroup of MPCs are labeled by *Nestin*-GFP, to test our hypothesis. We found increased expression of ubiquitin E3 ligases and ubiquitinated proteins in callus tissues at the early phase of fracture repair. Proteasome inhibitor Bortezomib, given soon after fracture, enhanced fracture repair, which is accompanied by increased callus *Nestin*-GFP+ cells and their proliferation, and the expression of osteoblast-associated genes and Runx2 and JunB proteins. Thus, early treatment of fractures with Bortezomib could enhance the fracture repair by increasing the number and proliferation of MPCs.

## Introduction

Successful fracture repair restores broken bones to their original structural geometry and biomechanical integrity. The fracture repair process is comprised of four phases: inflammatory response; recruitment of mesenchymal progenitor/stem cells (MPCs) and their differentiation into osteoblasts (OBs); new bone formation; and remodeling of the new bone from woven to lamellar bone. Although fracture repair is rapid and efficient, about 5–10% of patients with fractures, mainly aged subjects, develop delayed union or nonunion, resulting in unsuccessful fracture

**Funding:** This work was supported by grants from National Institute of Health PHS awards (P30AR061307pilot to H.Z., R01AG059775 to L. X.), NYSTEM N13G-084 (C029548 to L.X.), and National Natural Science Foundation of China (No.81673769 to L.P., No.31771560 to H.Z.). H.Z. is responsible for conception and design, material collection, data collection, data analysis, data interpretation, manuscript writing and final approval of manuscript; L.P. is responsible for data interpretation and final approval of manuscript; L.X. is responsible for conception and design, financial support, manuscript writing, and final approval of manuscript. The funders had no role in study design, data collection and analysis, decision to publish, or preparation of the manuscript.

**Competing interests:** NO authors have competing interests.

repair [1, 2]. BMP7 and PTH are FDA-approved drugs for the treatment of nonunion of long-bone fractures [3, 4]. However, short half-life, side effects, and high cost limit their utilization.

The Ubiquitin/Proteasome System (UPS) is a major mechanism for regulating the ubiquitination and degradation of intracellular proteins, including transcription factors or abnormal proteins trafficking from the cytosol and nuclei [5]. Disruption of the UPS has been linked to pathophysiological conditions that provoke accumulation of aberrant proteins, such as in neurodegenerative disorders and cancers [6]. Ubiquitination is carried out via sequential enzymatic reactions involving ubiquitin activating enzyme E1, ubiquitin-conjugating enzyme E2, and ubiquitin ligase E3, which confer substrate specificity, linking the ubiquitin-protein conjugation machinery to target molecules. Ubiquitinated (Ub-) proteins often undergo degradation in proteasomes or lysosomes, resulting in decreased steady-state levels of one or a group of proteins. A number of positive regulators of OBs, including the BMP- [7] & TGFβ-Smads [8], Runx2 [9], and JunB [10–13], can be regulated via the UPS by Nedd4 sub-class of E3 ligases, which comprises 7 mammalian members, including Nedd4-1, Nedd4-2, Itch, Wwp1&2, and Smurf 1&2. We have investigated the role of Smurf1, Wwp1 and Itch in bone using Smurf1−/−, Wwp1−/−, and Itch−/− mice and demonstrated that all of them negatively regulate OB differentiation or migration by controlling the stability of Runx2, JunB, or CXCR4 proteins in OBs and MPCs [10–16]. However, the involvement of these E3s and their effects on protein ubiquitination and degradation in fracture repair has not been well studied.

Proteasome inhibitors have been reported to induce OB differentiation [17]. Local delivery of the proteasome inhibitor, PS1 (Z-Ile-Glu(OtBu)-Ala-Leu-CHO) promotes femoral closed fracture repair in rats, accompanied by increasing BMP2 expression in callus [18]. BMP2 has strong osteoinductive activity to regulate OB differentiation and has been used to accelerate fracture and wound healing in patients [19]. Bortezomib (Btz) and Carfilzomib are two FDA-approved proteasome inhibitors for treating patients with multiple myeloma. Systemic administration of Btz enhances bone formation in mice by stabilizing Runx2 protein in MPCs and promoting their differentiation into OBs [20]. However, long-term use of proteasome inhibitors also inhibits osteoclast differentiation by inhibiting NF-κB signaling [21] during the bone remodeling stage of fracture repair, which may be counteractive to their bone anabolic effects on OBs. Thus, whether proteasome inhibitors can promote fracture repair by targeting only MPCs has not been investigated. We hypothesize that fracture injury triggers the UPS in callus at the early phase of fracture repair, leading to elevated protein ubiquitination and degradation, including Runx2 and JunB, and decreased MPCs. Btz, given only at the early phase of fracture repair, could promote fracture repair by increasing MPCs, OB differentiation, and Runx2 and JunB expression.

In this study, we used a tibial fracture model in *Nestin*-GFP reporter mice [22] in which a subgroup of MPCs are *Nestin*-GFP positive [23]. We examined the expression pattern of the Nedd4 sub-class of E3s and Ub-proteins in fracture callus and the effect of Btz on fracture repair focusing on *Nestin*-GFP+ MPCs.

## Materials and methods

### Animal study

**Animals and ethics statement.** *Nestin*-GFP transgenic mice (*Nestin*-GFP mice) were obtained from Dr. Tatyana V Michurina (Cold Spring Harbor Laboratory) [23] and generated on a C57BL/6J background, in which the 5.8-kb fragment of the promoter region was inserted upstream of sequences encoding destabilized eGFP. Primers: Forward: 5′-ATCACATGG TCCTGCTGGAGTTC-3′, Reverse: 5′-GGAGCTGCACACAACCCATTGCC-3′ were used for genotyping *Nestin*-GFP mice. Three-month-old C57BL/6J wild type (WT) mice were

purchased from the National Cancer Institute. Male mice were used in this study to avoid any potential influence of estrogen on bone. Mice were randomly grouped (N = 3–8 mice /group or time point). Mice were housed in conditions of specific pathogen-free and controlled temperature (22˚C) and illumination (12-hour light/12-hour dark cycle). All animal procedures were conducted in accordance with approved guidelines of the University of Rochester Committee for Animal Resources (protocol number: 2001-121R).

**Tibial fracture model, Bortezomib treatment, and callus preparations.** For fracture model, open tibial fracture procedure is a standard operating procedure (SOP) used in the Center for Musculoskeletal Research (CMSR) [24–27]. Briefly, a 6 mm long incision is made in the skin over the anterior tibia after anesthesia (Ketamine 100mg/kg and Xylazine 10 mg/kg, intraperitoneal injection). A sterile 27G needle is inserted via the proximal tibial articular surface into the bone marrow (BM) cavity, temporarily withdrawn to facilitate midshaft transection of the tibia using a scalpel, and reinserted to stabilize the fracture, which is confirmed by X-ray. The incision is closed with sutures. Mice are given Buprenorphine SR (ZooPharm) 0.5 mg/kg subcutaneously 4 hours before the surgery to control pain, and are monitored every 24 hours for 72 hours post-fracture to ensure that mice are under a pain-free condition. For Btz treatment, Btz is purchased from Cambridge, MA, USA, and dissolved in saline at the concentration of 0.1mg/ml. Mice are given Btz (0.6 mg/kg body weight) or vehicle by intraperitoneal (i.p.) injection on 1-, 3- and 5- day post fracture (dpf). Mice are euthanized by $CO_2$ inhalation and secondary cervical dislocation at different time points. For callus cell preparation, callus is harvested at 10 dpf and dissected free of soft tissue and cut into pieces. Callus pieces are washed thoroughly with cold PBS and then digested with ACCUMAX cell detachment solution (Stem cell Tech) for 30 minutes at room temperature. $8 \times 10^5$–$1.5 \times 10^6$ cells are obtained from each mouse (two fractures).

**Micro-CT and Bio-mechanical testing.** For microCT, fractured tibiae at 10 dpf were dissected free of soft tissue, fixed overnight in 70% ethanol and removed the stabilizing needle before scanning at high resolution (10.5 μm) on a VivaCT40 micro-CT scanner (Scanco Medical, Basserdorf, Switzerland) using 300 ms integration time, 55kVp energy, and 145 uA intensity. 3-D images were generated using a constant threshold of 275 for all samples. For Bio-mechanical testing, fractured tibiae at 28 dpf are stored at −80˚C after removing the stabilizing needle carefully to avoid any damage to the architecture of the callus. The tibial ends are embedded in polymethylmethacrylate and placed on an EnduraTec system (Bose Corporation). A rotation rate of 1˚/s is used to twist the samples to failure or up to 80˚. Maximum torque, maximum rotation, and torsion rigidity are analyzed using a CMSR SOP [28].

## Cell analysis

**Flow cytometry and cell sorting.** APC-anti-CD45, PE-anti-CD105 and PE-cy7-anti-Sca1 antibodies (Abs) are purchased from eBioscience. Callus and BM cells are stained with various fluorescein-labeled Abs and subjected to flow cytometric analysis using a Becton-Dickinson FACSCanto II Cytometer, according to the manufacturer's instructions. Results are analyzed by Flowjo7 data analysis software (FLOWJO, LLC Ashland, OR). For cell sorting, BM cells from *Nestin*-GFP mice are sorted using a Becton-Dickinson FACSAria III sorter, according to the manufacture's instructions [29].

**Cell proliferation and apoptosis assays.** For proliferation assays, *Nestin*-GFP mice are given two i.p. injections of BrdU (1 mg/mouse/injection) spaced 16 hours apart, and are euthanized 2 hours after the second injection. Callus cells are harvested and stained with APC-conjugated anti-BrdU Ab for flow cytometry using a BrdU Flow Kit (BD Pharmingen, cat#: 51-2354AK), according to the manufacturer's instructions. The percentage of BrdU

+ cells in the *Nestin*-GFP+ cells is assessed. For apoptosis assays, cells are stained with APC-anti-AnnexinV using an Apoptosis Detection Kit (eBioscience, cat#: 88–8007,). After being stained with Propidium iodide (PI), the percentage of PI-/AnnexinV+ apoptotic cells is assessed by flow cytometry. Of notes, some of terminally differentiated OBs may be Annexin V positive, which cannot be excluded by this method [30].

## Histological analysis

**Histology and histomorphometric analysis.** Tibiae are fixed in 10% buffered formalin, decalcified in 10% EDTA, and paraffin sections are prepared for histology based on SOP of CMSR [31, 32]. In brief, bones are embedded in paraffin and 4 μm thick sections are cut at 3 levels. Sections are then stained with Alcian Blue/Hematoxylin (ABH) and counterstained with Eosin/Orange G. Sections are also stained for Tartrate-resistant acid phosphatase (TRAP) activity for osteoclast identification. Stained sections are scanned with an Olympus VS-120 whole-slide imaging system. Histomorphometric measurement of cartilage, woven bone area and osteoclast numbers is performed on 2–3 sections/bone using Visiopharm software (Version 2018.4) blindly, as we previously described [31, 32].

**Immunostaining.** Bones are embedded in Tissue-Tek and sectioned at 8 μm using a Leica CM1850 cryostat (Leica, German). Unstained sections are directly observed (below) or subjected to immunofluorescence staining after blocking in PBS with 10% normal goat serum and 0.2% Triton X-100 for 1 hour and staining overnight with rat anti-mouse Endomucin Ab (1:50, Santa Curz, cat#: sc-53941), mouse anti-Ubiquitin Ab (1:50, Santa Curz, cat#: sc-166553) or mouse anti-BrdU Ab (1:100, Santa Curz, cat#: sc-32323) at 4˚C. After rinsing with PBS for 15 minutes, sections are incubated with goat anti-rat Alexa Fluor 568 (1:400, Invitrogen, cat#: A-11011) or goat anti-mouse Alexa Fluor 568 (1:400, Invitrogen, cat#: A-11004) at room temperature. Unstained and stained slides are mounted with mounting medium containing DAPI (Vector). Stained sections are scanned with an Olympus VS-120 whole-slide imaging system. Scanned images are numbered and the external callus area is outlined. The percentage of positively stained cells over the total cells within the external callus is quantified by Image J software (Version 1.53).

## Biochemical analysis

**Quantitative real time RT-PCR.** Fracture callus tissues are homogenized under liquid nitrogen. Total RNA is extracted using TRIzol reagent. cDNAs are synthesized using an iSCRIPT cDNA Synthesis Kit (Bio-Rad). Quantitative RT-PCR amplification is performed in an iCycler (Bio-Rad) real time PCR machine using an iQ SYBR Green supermix (Bio-Rad), according to the manufacturer's instructions. *β-actin* is amplified on the same plates and used to normalize the data. Each sample is prepared in triplicate and each experiment is repeated at least three times. The relative abundance of each gene is calculated by subtracting the CT value of each sample for an individual gene from the corresponding CT value of *β-actin* (ΔCT). ΔΔCT are obtained by subtracting the ΔCT of the reference point. These values are then raised to the power 2 (2ΔΔCT) to yield fold-expression relative to the reference point. Representative data are presented as means ± SD of the triplicates or individual samples. The sequences of primers and qPCR conditions in the current study are shown in S1 Table.

**Western blotting analysis.** Fracture calluses are homogenized under liquid nitrogen. Proteins are extracted with RIPA lysis buffer. Proteins are quantitated using a kit from Bio-Rad and loaded in 10% SDS-PAGE gels and blotted with anti-Ubiquitin, JunB, Runx2, NF-κB RelA and β-actin (Santa Cruz), IκBα and β-catenin (Cell Signaling), Osteocalcin (Enzo ALX) Abs as we described previously [11]. Bands are visualized using ECL chemiluminescence (Bio-rad).

## Statistical analysis

All results are given as mean ± SD. Statistical analysis is performed using GraphPad Prism 5 software (GraphPad Software Inc., San Diego, CA, USA). Comparisons between 2 groups are analyzed using a 2-tailed unpaired Student's t test. One-way ANOVA and Tukey's post-hoc multiple comparisons are used for comparisons among 3 or more groups. p values <0.05 are considered statistically significant. The sample size for biomechanical testing is determined based on our previous published data using mice with tibial fracture, in which callus samples were harvested at 28 dpf following drug treatments [33]. The sample size n> = 3 is calculated given expected means of torsional rigidity 1491 ± 58 [N.mm/(rad/mm)] in vehicle-treated mice, 639 ± 133 [N.mm/(rad/mm)] in drug 1-treated mice, and 1141 ± 197 [N.mm/(rad/mm)] in drug 2-treated mice, with an alpha level of 0.05, and power of 80%, for One-way ANOVA with Dunnett's test.

## Results

### Increased expression of Nedd4 sub-class E3 ligases and ubiquitinated proteins in fracture callus

We and others have reported that deletion of Nedd4 sub-class of E3 ligases promotes OB differentiation by preventing degradation of OB positive regulators, Runx2 [9] and JunB [13]. To determine if E3-mediated ubiquitination and degradation of these proteins play a role in fracture repair, we examined the expression levels of Nedd4 sub-class of E3s, *Wwp1*, *Itch*, *Smurf1* and *Smurf2*, in callus samples at different time points after fracture by qPCR. Compared to samples isolated from time 0, the expression of *Wwp1*, *Itch*, and *Smurf2* increased significantly, starting from 4 dpf, peaked at 7 dpf, and returned to baseline at 14 dpf and 21 dpf. *Smurf1* expression increased only mildly during the whole fracture repair process, while *Itch* expression increased more than the others (Fig 1A). We examined levels of total Ub-proteins at 1 hour, 4 and 7 dpf when E3 ligase levels were increased. We found that the expression levels of Ub-proteins in callus were markedly increased compared to the time 0 sample and to non-fractured bones at all time points. In contract, total Ub-proteins in non-fractured bones were no change (Fig 1B).

### Bortezomib increases fracture repair

Because E3 ligases cause ubiquitination of target proteins that subsequently undergo proteasomal degradation and Ub-proteins are increased in the early phase of fracture repair (Fig 1), we want to know if short-term of proteasome inhibition at the early phase of fracture repair could enhance fracture repair. We treated mice with Btz, a proteasome inhibitor used clinically, at 1-, 3- and 5-dpf and sacrificed them at different time points. We used Btz at 0.6 mg/kg body weight, a dose for treating multiple myeloma and increasing bone volume in mice [34]. ABH-stained sections revealed that Btz significantly increased cartilage and woven bone area at 10 and 14 dpf (Fig 2A & 2B), which was confirmed by micro-CT analysis (Fig 3A). Btz increased the mRNA expression of genes related to OB differentiation (*Osx*, *Runx2*) and maturation (*Bsp*, *Ocn*, *Opn*), and chondrocyte differentiation (*Sox9*, *Col2*) in callus (Fig 3B). Increased OB function was further demonstrated by increased Osteocalcin protein levels in callus samples from Btz-treated mice (Fig 3C). TRAP-stained sections showed that Btz slightly decreased osteoclast numbers at 10 dpf, but not in other time points (Fig 4A & 4B). Bio-mechanical testing showed increased torsional rigidity and maximum Torque in Btz-treated mic at 28 dpf (Fig 4C). Osteoclastogenesis is controlled by NF-κB/NFATc1 signal pathway in osteoclast precursors [35]. We examined the expression levels of *nf-κb* and *nfatc1* mRNA in callus at 10 and

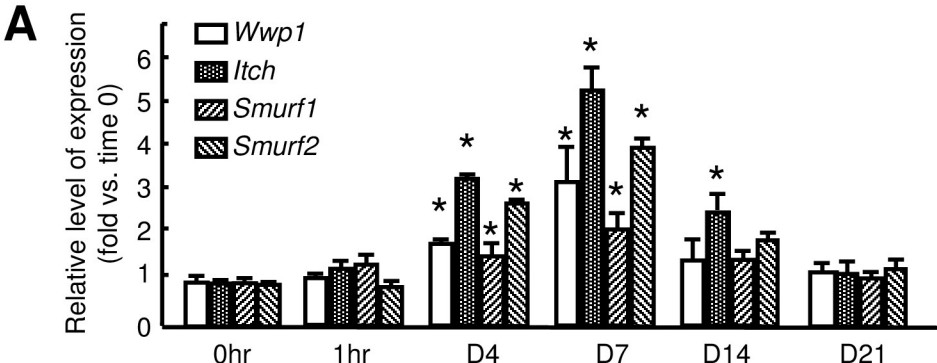

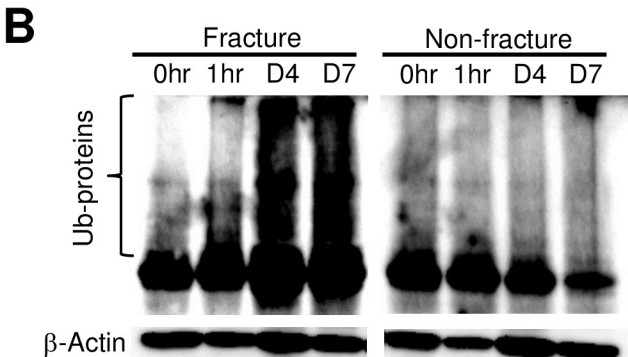

**Fig 1. Increased expression of E3 ligases and ubiquitinated proteins in fracture callus.** 3-month-old C57BL/6J male mice received an open tibial fracture surgery. (A) Expression levels of E3 ligases in callus tissues determined by qPCR at different time points. N = 3 mice. Values are means ± SD. Relative mRNA expression is the fold-change vs. the time 0 samples as 1. *p<0.05 vs. time 0 samples. (B) Total Ub-proteins in callus tissues determined by Western blot analysis using anti-Ubiquitin Ab at different time points. N = 3 mice. Experiments were repeated independently at least 2 times.

21 dpf and found that Btz significantly reduced *nf-κb*, and *nfatc1* expression at 10, but not at 21 dpf (Fig 4D). Of note, Btz did not reduce the expression of *Rankl* in the same samples, suggesting that the inhibitory effect of Btz in osteoclasts is likely due to reduced expression of osteoclast transcription factors, *nf-κb* and *nfatc1*.

### Increased *Nestin*-GFP+ mesenchymal progenitor/stem cells in callus during bone fracture repair

In WT mice, Btz enhances bone formation in mice by preventing Runx2 degradation in BM MPCs [20]. To study if Btz-enhanced fracture repair involves the similar cell population in callus, we utilized *Nestin*-GFP reporter mice, which allows us to detect *Nestin*-GFP+ cells in callus by flow cytometry and fluorescent microscope. We used CD45-CD105+Sca1+ as surface markers for MPCs and demonstrated that in BM of non-fractured mice, less than 0.1% of cells were *Nestin*-GFP+ cells. Among them 80% of *Nestin*-GFP+ cells were CD105+Sca1+ and 74% of CD105+Sca1+ cells were *Nestin*-GFP+ (Fig 5A). Correlation analysis revealed a significant association between *Nestin*-GFP+ cells and CD45-CD105+Sca1+ cells ($R^2$ = 0.884) (Fig 5B). In fracture callus, the percentage of *Nestin*-GFP+ cells increased at the early phase of fracture repair. Compared to time 0 samples, the time point when less than 0.1% of cells were *Nestin*-GFP+ cells, *Nestin*-GFP+ cells markedly increased at 7 dpf, peaked at 10

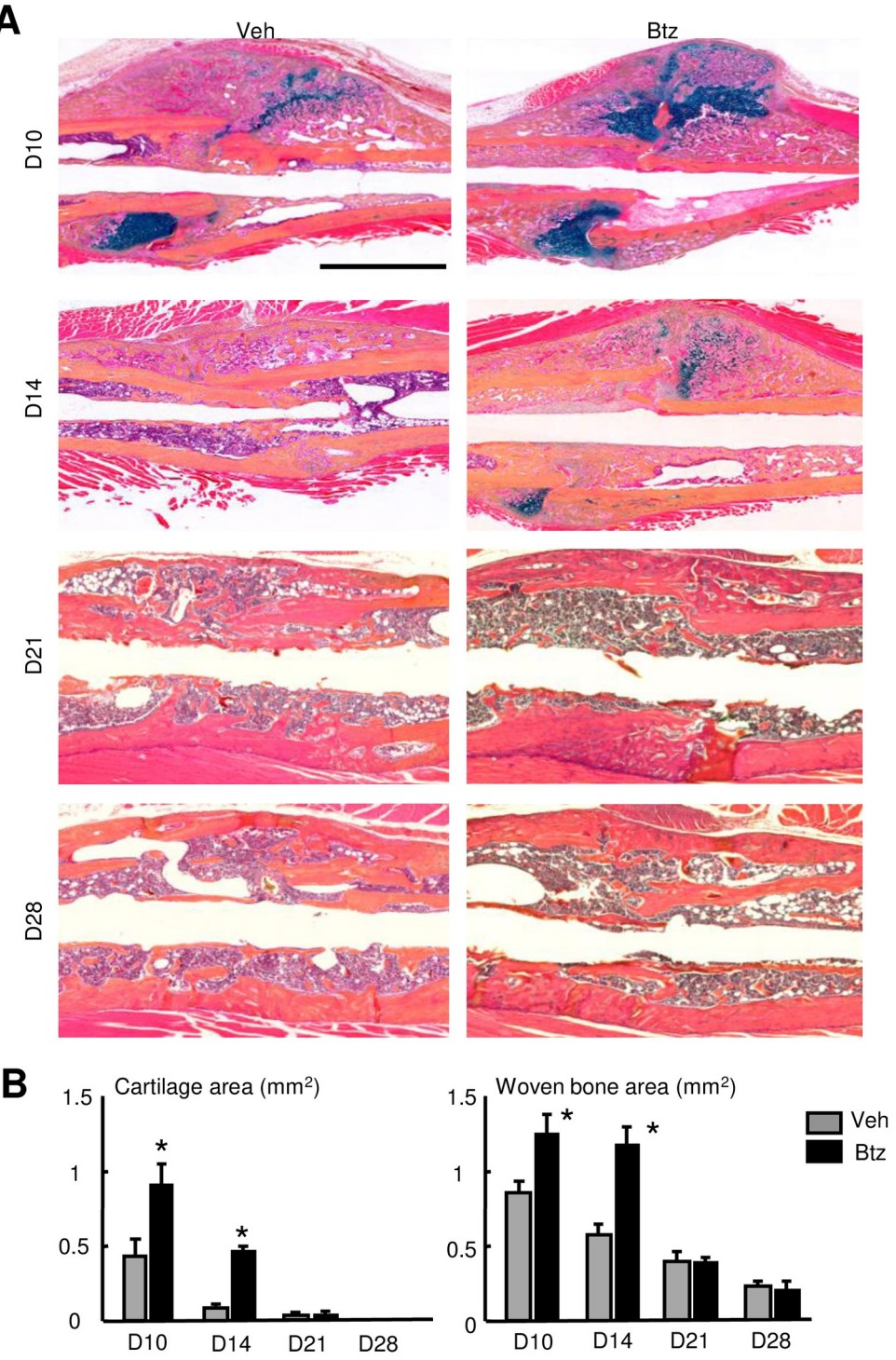

**Fig 2. Bortezomib increases cartilage and woven bone areas in fracture callus.** 3-month-old C57BL/6J male mice received fracture surgery as in Fig 1. Mice were treated with Btz (0.6 mg/kg body weight, i.p.) on 1-, 3- and 5-dpf, and were sacrificed at different times. (A) Representative ABH-stained sections showing higher woven bone and callus areas in Btz-treated mice. Scale bar = 1 mm. (B) Woven bone and cartilage areas analyzed by Image J software. N = 8 mice. Values are means ± SD. *p< 0.05 vs. vehicle-treated mice.

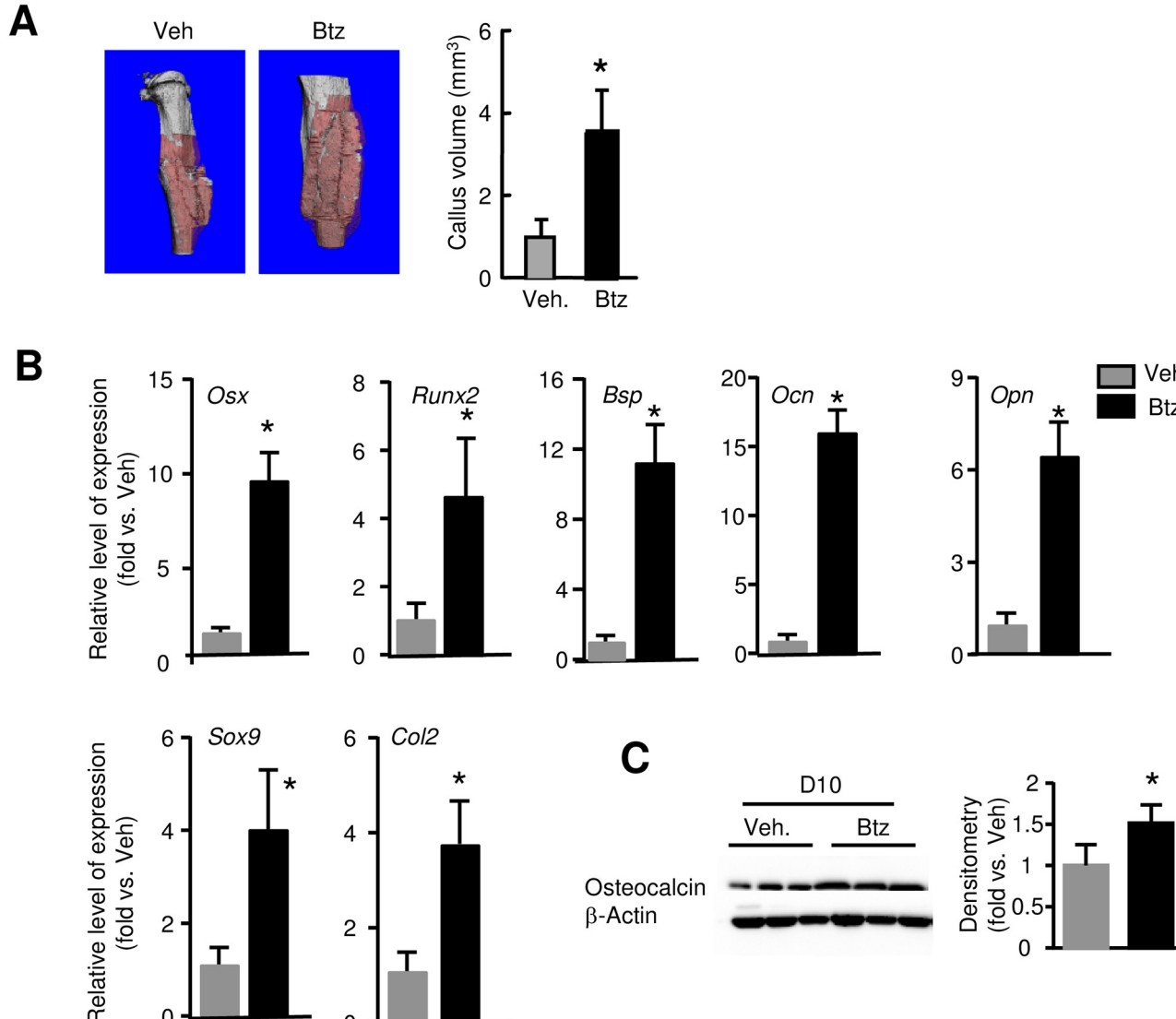

**Fig 3. Bortezomib increases callus volume and the expression of genes related to osteoblast and chondrocyte differentiation in fracture callus.**
3-month-old C57BL/6J male mice received fracture surgery and treatment as in Fig 2. (A) Callus volume measured at 10 dpf by micro-CT. N = 8 mice. Values are means ± SD. *p< 0.05 vs. vehicle-treated mice. (B) Expression levels of genes related to OB differentiation and maturation, and chondrocyte differentiation in callus tissues examined by qPCR at 10 dpf. N = 3 mice. Values are means ± SD. Relative mRNA expression is the fold-change vs. vehicle-treated mice as 1. *p< 0.05 vs. vehicle-treated mice. (C) Expression levels of Osteocalcin in callus tissues following Btz treatment at 10 dpf examined by Western blot analysis. Data is normalized to callus samples from vehicle-treated mice. N = 3 mice. *p< 0.05 vs. vehicle-treated mice.

dpf, and returned to very low levels at 21 dpf (Fig 5C). *Nestin*-GFP+ cells can give rise to pericytes during the bone repair process in a calvarial defect model [36]. We examined the association between *Nestin*-GFP+ cells and blood vessels in callus sections by immunofluorescence staining with anti-Endomucin Ab (Fig 5D) and found that 34% of *Nestin*-GFP + cells were located adjacent to Endomucin+ blood vessels (Fig 5E).

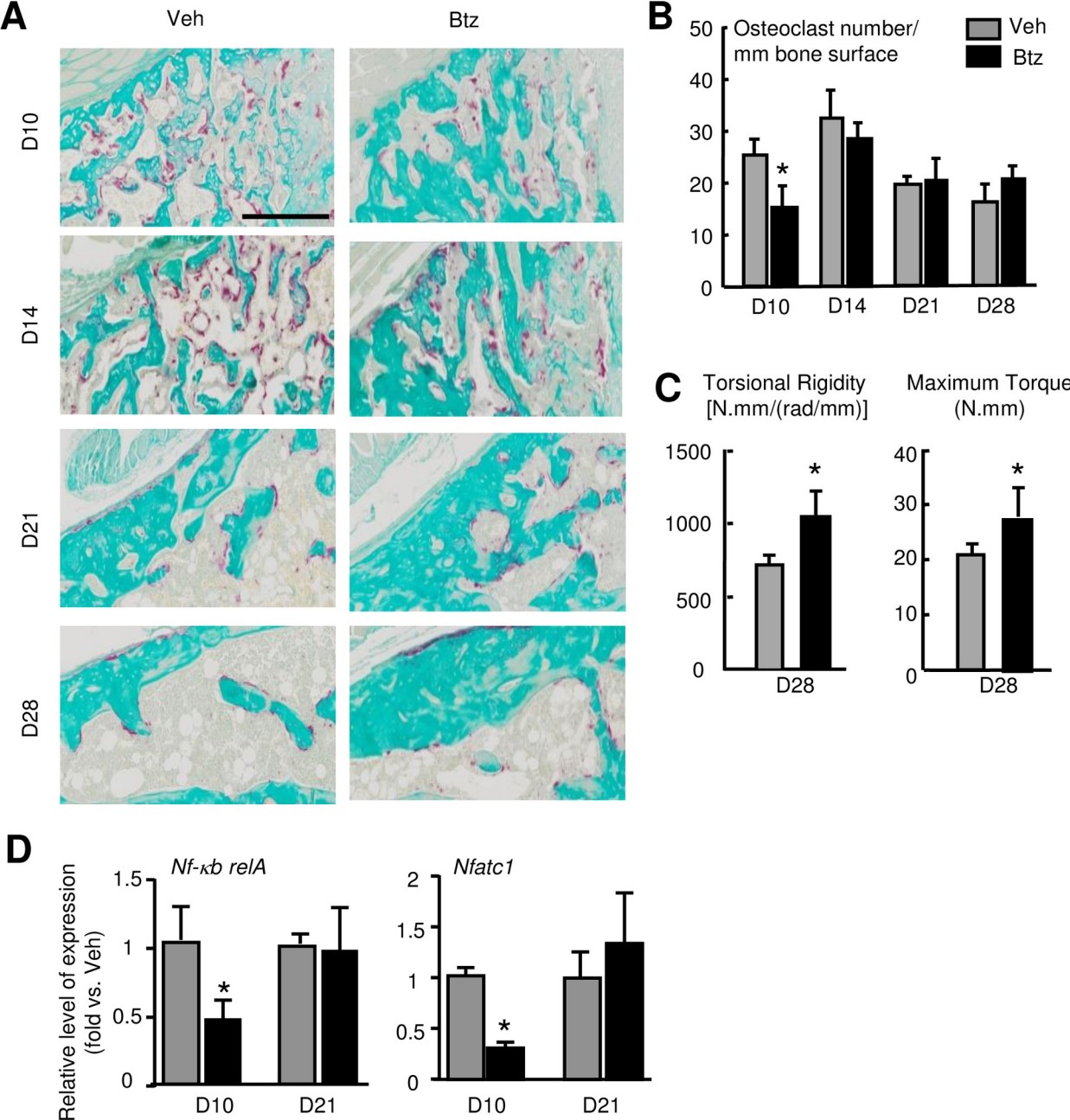

**Fig 4. Bortezomib decreases osteoclasts and improves fracture repair in mice.** 3-month-old C57BL/6J male mice received fracture surgery were treated with Btz as in Fig 2. (A) Representative TRAP-stained sections showing fewer TRAP+ osteoclasts at 10 dpf in Btz-treated mice. Scale bar = 0.25 mm. (B) TRAP+ osteoclasts in callus analyzed by Image J software. N = 8 mice. Values are means ± SD. *p< 0.05 vs. vehicle-treated mice. (C) Bone stiffness and strength assessed by biomechanical testing at 28 dpf. N = 5 mice. Values are means ± SD. *p< 0.05 vs. vehicle-treated mice. (D) Expression of osteoclast transcription factors, *nf-kb* and *nfatc1* examined by qPCR at 10 dpf. N = 3. Values are means ± SD. Relative mRNA expression is the fold-change vs. samples from the vehicle-treated mice as 1. Unpaired t-test is used to compare Veh and Btz treated day 10 and day 21 callus, respectively. *p< 0.05 vs. vehicle-treated mice.

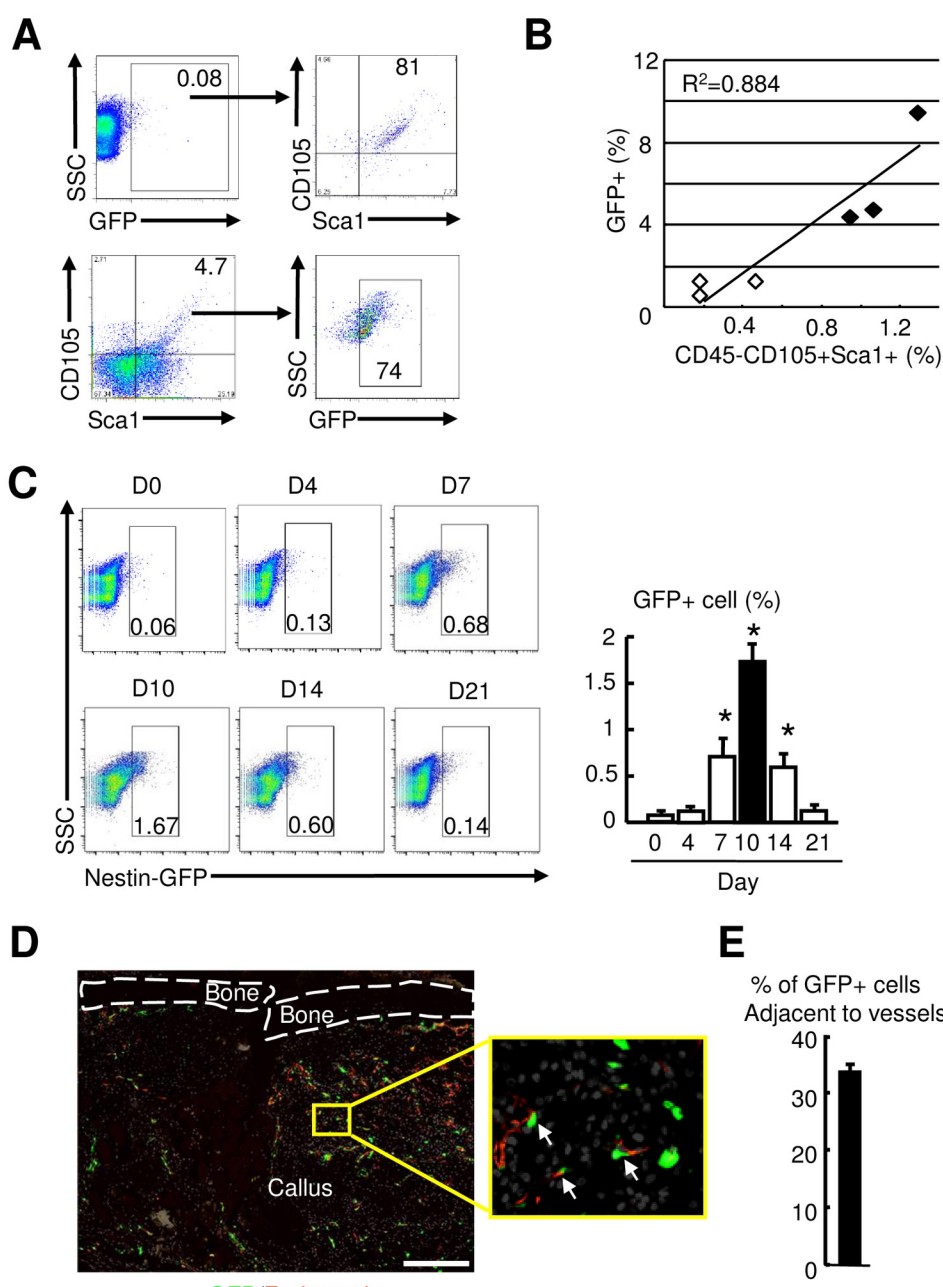

**Fig 5. Increased *Nestin*-GFP+ mesenchymal stem/progenitor cells in callus at the early phase of fracture repair.**
3-month-old *Nestin*-GFP male mice were used. (A) cells were stained with anti-CD105 and Sca1 Abs and subjected to flow cytometry. GFP+ (upper) or CD105+Sca1+ cells were gated for analysis. N = 5 mice. SSC: side-scattered light. (B) Correlation of *Nestin*-GFP+ and CD45-CD105+Sca1+ cells. (C) The percentage of *Nestin*-GFP+ cells in callus tissues analyzed by flow cytometry. N = 3. Values are means ± SD. *p< 0.05 vs. time 0 samples. (D) Frozen sections subjected to immunofluorescence stained with anti-Endomucin Ab for blood vessels. The insert indicates enlarged immunofluorescence stained image showing *Nestin*-GFP+ cells adjacent to Endomucin+ blood vessels (arrows). Scale bar = 1 mm. (E) The *Nestin*-GFP+ cells adjacent to Endomucin+ blood vessels (distance < 2 cell diameter to blood vessel) counted. The percentage of *Nestin*-GFP+ cells adjacent to blood vessels over the total number *Nestin*-GFP + cells calculated. N = 3 mice. Values are means ± SD.

## Bortezomib increases *Nestin*-GFP+ cells, ubiquitinated proteins, and proliferation of *Nestin*-GFP+ cells in fracture callus

To determine if Btz increases MPCs in fracture callus, we treated *Nestin*-GFP mice with Btz and examined the frequency of GFP+, CD105+ or Sca1+ cells by flow cytometry using callus samples harvested at 10 dpf. Compared to vehicle-treated mice, Btz-treated mice had significantly more *Nestin*-GFP+ cells (1.2±0.9% vs. 9.6±0.2%), CD105+ cells (4.1±0.8% vs. 8.9±0.4%), Sca1+ cells (4±1.5% vs. 9±0.5%), and CD45-CD105+Sca1+ cells (0.3±1.1% vs. 1.4±0.1%) (Fig 6A), but not B220+ B cells, CD3+ T cells and CD11b+ monocytes (Fig 6B). Western blotting analysis showed more Ub-proteins and higher expression levels of Runx2 and JunB proteins in callus of Btz-treated mice (Fig 6C).

To determine the distribution of Ub-proteins, we performed immunofluorescence staining with an anti-Ubiquitin Ab on callus sections. Btz markedly increased the Ubiquitin-positive stained areas and cells that were both *Nestin*-GFP and Ubiquitin positive, which were located mainly in the callus (Fig 7A & 7B). To study the potential mechanisms that mediate Btz-induced increase in *Nestin*-GFP+ cells, we examined cell proliferation by BrdU incorporation assays at 10 dpf, a time when increased *Nestin*-GFP+ cells in callus peaked. Immunofluorescence staining with an anti-BrdU Ab revealed more *Nestin*-GFP+BrdU+ cells in callus sections from Btz-treated mice than those from vehicle-treated mice (Fig 7C & 7D). Flow cytometric analysis indicated that Btz increased the percentage of *Nestin*-GFP+BrdU+ cells by 2-fold (21.7±4.8% in Btz- vs. 11.2±2.1% in vhecile-treated mice, Fig 7E). In contrast, Btz did not affect AnnexinV+ apoptotic cell numbers (Fig 7F). These data indicate that Btz increases MPCs and protein ubiquitination in fracture callus. Btz increases proliferation of *Nestin*-GFP+, which may contribute to increasing their numbers in fracture callus.

## Bortezomib increases β-catenin and inhibits NF-κB protein expression in fracture callus

β-catenin and NF-κB signal pathways play critical roles in controlling osteoblastogenesis and osteoclastogenesis, respectively, and both of them are regulated by the UPS [21]. To examine if Btz promotes fracture repair by affecting β-catenin and NF-κB signal proteins, we examined the expression levels of β-catenin, NF-κB inhibitor IκBα, and NF-κB RelA proteins in callus tissues by Western blot analysis. We found that Btz increased the expression of IκBα and β-catenin on 10 dpf, but not 21 dpf, the time when Btz's effect has disappeared. Accordingly, NF-κB-RelA levels were decreased at 10 dpf in Btz-treated mice (Fig 8).

## Discussion

Protein ubiquitination and degradation mediated by the UPS plays an important role in bone cell regulation under normal and pathological conditions. The expression levels of a number of positive regulators in MPCs are tightly controlled by their ubiquitination status. Proteasome inhibitors promote OB differentiation and bone formation by reducing the degradation of ubiquitinated OB positive regulators [7, 8, 20, 37–41]. Bone fracture repair requires proliferation and OB differentiation of MPCs. Proteasome inhibitors enhanced fracture repair in mouse models of bone defects [42]. Here, we demonstrated that total ubiquitinated protein levels in fracture callus increases during the early stages of fracture repair. Short-term administration of the proteasome inhibitor, Btz in fractured *Nestin*-GFP mice enhanced fracture repair, which is associated with increased proliferation of *Nestin*-GFP+ cells, CD45-CD105+Sca1+ MPCs and expression of ubiquitinated proteins and positive OB regulators, Runx2 and JunB in callus.

Thus, UPS-mediated cellular processes may play a role in fracture repair, which involves in regulating proliferation of MPCs.

Our data indicate that the majority of protein ubiquitination and degradation happens early during fracture repair, associated with an increased expression of the Nedd4 sub-class of E3 ligases. We and others have reported that pro-inflammatory cytokines, such as TNF,

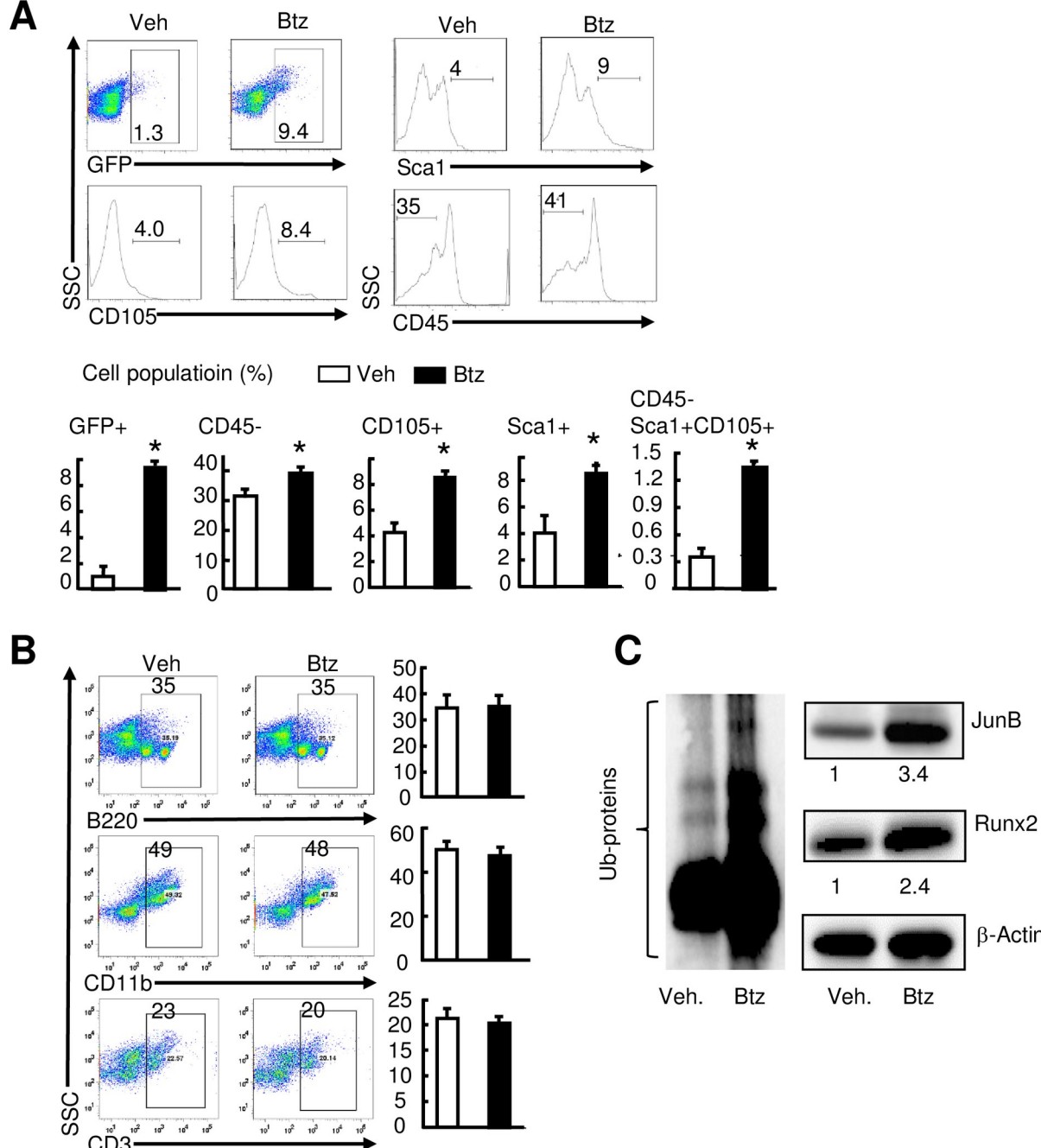

**Fig 6. Bortezomib increases the number of *Nestin*-GFP+ cells and expression of osteogenic proteins in fracture callus.** 3-month-old *Nestin*-GFP male mice received fracture surgery were treated with Btz as in Fig 2. Callus tissues were harvested at 10 dpf. The percentage of mesenchymal (A) and hematopoietic (B) lineage cell populations analyzed by flow cytometry. N = 4 mice. Values are means ± SD. *p< 0.05 vs. vehicle-treated mice. (C) Expression levels of total Ub-proteins, Runx2, and JunB examined by Western blot analysis. Numbers are the fold-change by densitometry analysis of Western blot images using callus samples harvested at 10 dpf from vehicle-treated mice as 1. N = 3 mice.

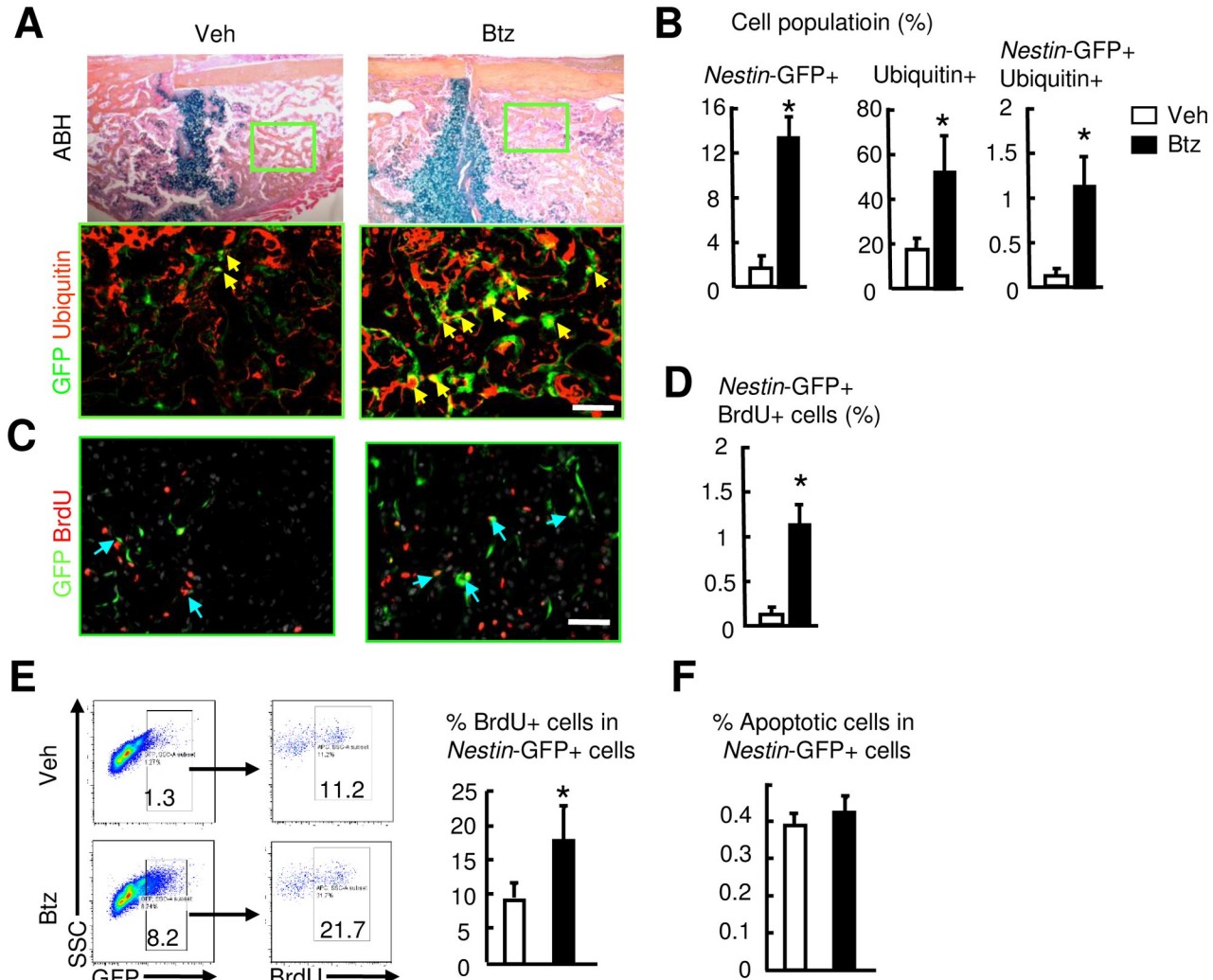

**Fig 7. Bortezomib increases the proliferation of *Nestin*-GFP+ cells in fracture callus.** 3-month-old *Nestin*-GFP male mice received fracture surgery were treated with Btz as in Fig 2. Callus tissues were harvested at 10 dpf. (A) Frozen sections subjected to immunofluorescence staining with anti-Ubiquitin Ab and examined under a fluorescent microscopy. *Nestin*-GFP+ cells are green and Ubiquitin+ cells are stained for red. *Nestin*-GFP+Ubiquitin+ cells are stained for yellow (indicated by yellow arrows). Scale bar = 1 mm. The adjacent sections stained for ABH. Inserts indicate enlarged immunofluorescence stained images. (B) The percentage of *Nestin*-GFP+, Ubiquitin+, or *Nestin*-GFP+Ubiquitin+ cells were counted. N = 3 mice. Values are means ± SD. *p< 0.05 vs. vehicle-treated mice. (C) Mice were given two injections of BrdU (1 mg/i.p. injection) spaced 16 hours apart, starting at 9 dpf. Frozen sections subjected to immunofluorescence staining with anti-BrdU Ab and analyzed under a fluorescent microscopy. *Nestin*-GFP+ cells are green and BrdU+ cells are stained for red. *Nestin*-GFP+BrdU+ cells are stained for yellow (indicated yellow arrows). Scale bar = 1 mm. (F) The percentage of *Nestin*-GFP+BrdU+ cells was counted. N = 3 mice. Values are means ± SD. *p< 0.05 vs. vehicle-treated mice. (D) The percentage of BrdU+ cells in callus *Nestin*-GFP+ cells analyzed by flow cytometry. N = 3 mice. Values are means ± SD. *p< 0.05 vs. vehicle-treated mice. (E) Cell apoptosis assessed by flow cytometry as the percentage of Annexin V+ cells in *Nestin*-GFP+ cells. N = 3 mice. Values are means ± SD.

increase transcription of E3 ligases via NF-κB, and that NF-κB binding elements are present in the promoter regions of Smurf1 [14] and Itch [43]. Cytokines, including IL-1, IL-6 and TNF, are highly expressed during the early inflammatory stage of fracture repair to initiate the repair cascade following injury [44, 45]. Thus, these pro-inflammatory cytokines may activate the UPS by up-regulating E3 ligases in fracture callus. Although depletion of a single E3, such as Smurf1 [16] or Itch [11], leads to enhanced bone formation by preventing the degradation of positive osteoblast regulatory proteins, we chose to use Btz in our present study because 1) Btz is an FDA-approved drug and has known bone anabolic effects, and 2) E3 ligases often have

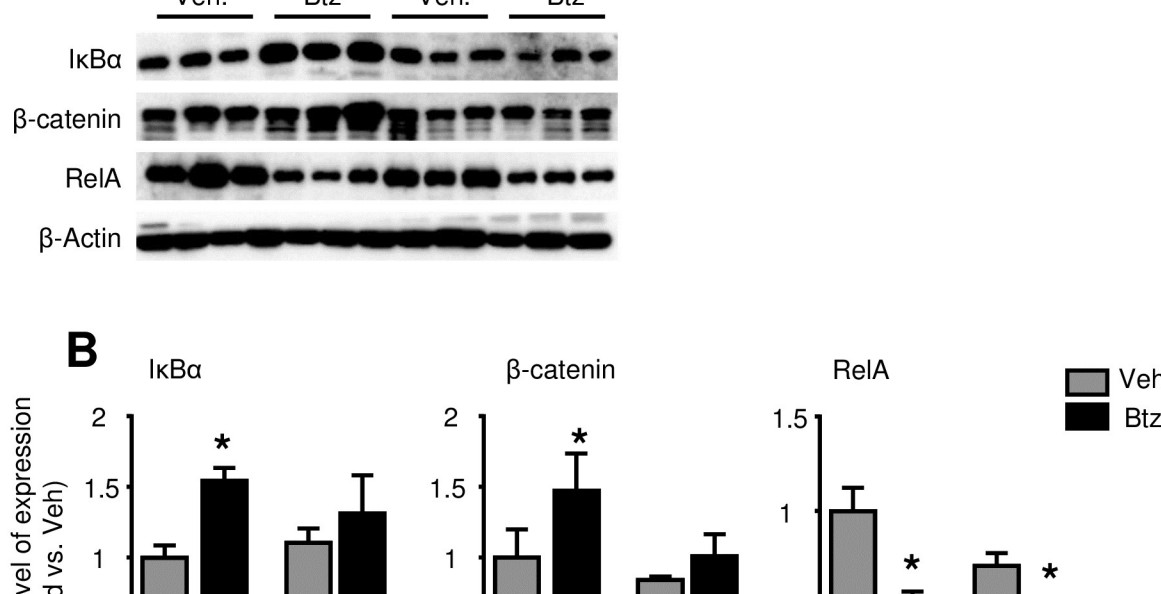

**Fig 8. Bortezomib increases IκBα, β-catenin and decreases RelA protein levels in fracture callus at 10-day post-fracture.** 3-month-old C57BL/6J mice received fracture surgery were treated with Btz as in Fig 2. Callus tissues were harvested at 10 and 21 dpf. (A) Expression levels of IκBα, β-catenin and RelA examined by Western blot analysis. (B) Densitometry analysis of Western blot images in A. Data is normalized to 10 dpf callus samples from vehicle-treated mice. N = 3 mice. Unpaired t-test is used to compare vehicle- and Btz-treated 10 dpf and 21 dpf callus, respectively. *p< 0.05 vs. vehicle-treated mice.

overlapping substrate proteins and vice versa. 3) We have successfully developed a bone targeted Btz (Bisphophonate conjugated Btz, BP-Btz) [33], which can facilitate the clinic use of Btz in bone fracture without the systemic side effect. Thus, targeting a single E3 ligase may not be efficient, especially since we have found levels of multiple E3 ligases elevated in fracture callus and Itch deficient mice have similar fracture repair as WT littermates (S1 Fig).

In this study, we used a short-term regimen for Btz, e.g. 3 administrations at early phase of fracture repair. The rationale why Btz was administered at day 1, 3, and 5 include 1) we hypothesize that Btz will increase MPCs at the early phase of fracture repair and day 1–5 is the time when MPCs are rapidly expanded [27, 46]; 2) a major mechanism of action of Btz is to block the degradation of Ub-proteins while we found increased Ub-proteins at the early phase of fracture repair (Fig 1B); and 3) our pharmacology study reported recently that blood Btz levels return to un-detectable levels at 12 hours [47] and that the inhibitory effect of Btz on Ub-protein degradation is abolished at 24 hours following a single Btz administration (manuscript in press). Treatment of mice with Btz on 1-, 3-, and 5-dpf enhanced fracture repair and bone strength (Figs 2 and 3), indicating that this short-term regimen is beneficial. At the cell level, Btz mainly increased cells in the mesenchymal lineage, but not other cell populations, such as B, T and monocytes at 10-dpf (Fig 3). Furthermore, BrdU incorporation assays revealed more BrdU+*Nestin*-GFP+ cells in Btz-treated mice, suggesting that Btz may increase proliferation of

this cell population. Because BrdU is able to participate the DNA repair as well [48], it is possible that Btz may prevent cells from entering apoptosis. This possibility can be studied in the future.

*Nestin*-GFP+ cells in *Nestin*-GFP mice have been used in stem cell studies with controversial results. Nestin was initially and widely used as the marker of neural stem cells [22]. Nestin + cells in BM include pericytes [36, 49], neural crest stem cells and mesenchymal stem cells [50]. Despite these controversial findings, our study for the first time using young adult mice indicated markedly increased *Nestin*-GFP+ cells in long bone fracture callus. We demonstrated that about 70% of *Nestin*-GFP+ cells are CD45-CD105+Sca1+, indicating that during the fracture healing, these *Nestin*-GFP+ cells are likely MPCs because they are rapidly expanded in response to injury as a recent study pointed out [46]. In addition, about 30% of *Nestin*-GFP+ cells are located adjacent to blood vessels, suggesting they are likely pericytes that to contribute to angiogenesis, another critical component of fracture repair.

One concern is the non-specificity of proteasome inhibitors because many proteins in various cell types can undergo ubiquitination in fracture callus. We believe that proteasome inhibitor promotes the fracture repair by increasing osteoblast differentiation through BMP2 [18]. Apart from this, our study strongly suggests that Btz increasing mesenchymal stem/progenitor cell is critical for fracture repair as well, particularly with a short-term regimen at early phase of fracture repair, and provides a strong rationale and evidence for the beneficial effects of short-term administration of proteasome inhibitors when they are given early during fracture repair when expansion of MPCs takes place. Short-term administration of proteasome inhibitors is unlikely to affect the OB differentiation in the whole repair process. Another issue is that fractures in young subjects typically heal well by themselves. Thus, it will be very important to determine if proteasome inhibitors could promote healing of delayed or nonunion fractures in aged or diabetic mice.

Another concern is the severe systemic side effect of Btz, which limits its use in clinic. Fortunately, we developed the BP-Btz, which was synthesized by conjugating Btz to a bisphosphonate with no antiresorptive activity. BP-Btz has significantly less systemic side than Btz, including thymic cell death, sympathetic nerve damage and thrombocytopenia. BP-Btz treatment prevented the bone loss in wild-type and ovariectomy mice and promoted the fracture healing in aged mice [33]. It will be also interesting to examine if newer generation of proteasome inhibitors with fewer side effects than Btz [51, 52], such as Carfilzomib, is efficacious with reduced systemic toxicity.

## Conclusion

Our study demonstrated elevated UPS in callus tissues during the early stages of fracture repair and the enhanced fracture healing by increasing MPCs via a short-term Btz administration. Our study provides the theoretical basis for treating fracture patients with proteasome inhibitors. Thus, manipulation of the UPS early after fracture may be a good strategy to promote bone fracture repair by proteasome inhibitors, particularly with our published bone targeting approach.

## Supporting information

**S1 Table. Primer sequences and qPCR conditions.**
(PPT)

**S1 Fig. Itch knockout mice have comparable bone fracture healing.** C57/B6 WT mice and Itch knockout out mice received open tibial fracture surgery and were sacrificed at 10, 14, 21,

and 28 days post fracture. Fractured tibiae examined by micro-CT. (A) Representative micro-CT images showing fracture callus. (B) Callus volume analyzed from micro-CT images. N = 3-5/time points.
(PPT)

**S1 File.**
(PPT)

## Author Contributions

**Conceptualization:** Hengwei Zhang, Xing Li, Lianping Xing.

**Data curation:** Hengwei Zhang, Xing Li, Jiatong Liu, Xi Lin.

**Formal analysis:** Xing Li, Jiatong Liu, Xi Lin.

**Funding acquisition:** Hengwei Zhang, Lingpeng Pei, Lianping Xing.

**Investigation:** Hengwei Zhang.

**Resources:** Xing Li.

**Supervision:** Lianping Xing.

**Writing – original draft:** Hengwei Zhang, Lianping Xing.

**Writing – review & editing:** Hengwei Zhang, Lingpeng Pei, Brendan F. Boyce, Lianping Xing.

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
