## [Decision Letter · Decision Letter 0]

22 Oct 2021

PONE-D-21-32074Proteasome Inhibition-Enhanced Fracture Repair is Associated with Increased Mesenchymal Progenitor Cells in MicePLOS ONE

Dear Dr. Liu,

Thank you for submitting your manuscript to PLOS ONE. After careful consideration, we feel that it has merit but does not fully meet PLOS ONE’s publication criteria as it currently stands. Therefore, we invite you to submit a revised version of the manuscript that addresses the points raised during the review process.

This manuscript must be considerably altered first of all for English then for the organization (Results) then other analyses must be performed as needed to give rise to robust results and conclusions, as specified by a referee and myself below.

We look forward to receiving your revised manuscript.

Kind regards,

Gianpaolo Papaccio, M.D., Ph.D.

Academic Editor

PLOS ONE

Journal Requirements:

2. To comply with PLOS ONE submissions requirements, in your Methods section, please provide additional information on the animal research and ensure you have included details on methods of anesthesia and/or analgesia.

7. Please include a copy of Table 1 which you refer to in your text on page 7.

Additional Editor Comments:

This study needs to be considerably amended for English. A native speaker expert in Scientific English must revise it in deep.

Results section should must be focused ONLY to the obtained results: is this the first manuscript they write?

Therefore please move all comments to the Discussion Section.

Authors must analyze several other osteogenic markers, such as BSP, OPN, OC, both for qRT-PCR and WB.

Better images related to colony formation in figure 4B MUST be added: they are not available.

Micro-Computed Tomography (μCT) and Radiological Analysis must also be added to increase the interest to this paper.

Reviewers' comments:

Reviewer's Responses to Questions

**Comments to the Author**

1. Is the manuscript technically sound, and do the data support the conclusions?

Reviewer #1: Partly

2. Has the statistical analysis been performed appropriately and rigorously? 

Reviewer #1: Yes

3. Have the authors made all data underlying the findings in their manuscript fully available?

Reviewer #1: Yes

4. Is the manuscript presented in an intelligible fashion and written in standard English?

Reviewer #1: No

5. Review Comments to the Author

Reviewer #1: In this study Authoirs examined the expression pattern of the Nedd4 sub-class of E3s and Ub proteins in fracture callus and the effect of Bortezomib on fracture repair on a tibial fracture model in Nestin-GFP reporter mice.

English should be revised. In particular, in paragraph: “Tibial fracture model, Bortezomib treatment, and callus preparations.” Authors used different time.

More references are needed in the first part of introduction.

“Bortezomib and Carfilzomib are 2 FDA-approved drugs”: please change 2 with two.

Results section should be revised. They should be reduced to results obtained. All the comments should be moved to the discussion section.

Authors should analyze other osteogenic markers, whose expression is related to that of Runx2, such as BSP, OPN, OC, both for qRT-PCR and WB.

Better images of colony formation in figure 4B should be provided.

Micro-Computed Tomography (μCT) and Radiological Analysis would add interest to the paper.

6. PLOS authors have the option to publish the peer review history of their article (what does this mean?). If published, this will include your full peer review and any attached files.

Reviewer #1: No

---

## [Author Response · Author response to Decision Letter 0]

13 Jan 2022

Jan 14, 2022

Responses to the journal staff

Re: PONE-D-21-32074R1

Thank you very much for working on our submission. 

1) We have included “Role of Funder statement: The funders had no role in study design, data collection and analysis, decision to publish, or preparation of the manuscript” in the cover letter.

2) In our previous submission, we have indicated in the cover letter that we have included “Supporting information files-uncut gels”. We ensured that we have uploaded this file and named it “Zhang uncut gels”.

3) In our previous submission, we have uploaded 3 Supporting Information files-1) Supporting-supplemental figure-Zhang S1 Fig; 2) Supporting-supplemental table-Zhang S1 Table; 3) supporting file-uncut gels-Zhang uncut gels. We ensured that we have uploaded these files. 

We have double checked our submission to ensure that we have uploaded all files.

Please contact us ASAP if you find there are any problems.

END

We have changed our manuscript to PLOS ONE's style requirements.

2. To comply with PLOS ONE submissions requirements, in your Methods section, please provide additional information on the animal research and ensure you have

included details on methods of anesthesia and/or analgesia.

We have followed the PLOS ONE submissions requirements. We have included the details on methods of anesthesia and/or analgesia (page 5)

We have made correction. 

4. PLOS ONE now requires that authors provide the original uncropped and unadjusted images underlying all blot or gel results reported in a submission’s figures or

Supporting Information files. This policy and the journal’s other requirements for blot/gel reporting and figure preparation are described in detail at

https://journals.plos.org/plosone/s/figures#loc-blot-and-gel-reporting-requirements and https://journals.plos.org/plosone/s/figures#loc-preparing-figures-from-imagefiles.

We have included the original uncropped and unadjusted images for Figure 1B, 2E, 5C, 7A in Supporting Information files #1-4. 

When you submit your revised manuscript, please ensure that your figures adhere fully to these guidelines and provide the original underlying images for all blot or gel data reported in your submission. See the following link for instructions on providing the original image data: https://journals.plos.org/plosone/s/figures#locoriginal-images-for-blots-and-gels.

In your cover letter, please note whether your blot/gel image data are in Supporting Information or posted at a public data repository, provide the repository URL if

relevant, and provide specific details as to which raw blot/gel images, if any, are not available. Email us at plosone@plos.org if you have any questions.

We have indicated in the cover letter that the original blot/gel image are included in Supporting Information files #1-4. 

5. We note that you have included the phrase “data not shown” in your manuscript. Unfortunately, this does not meet our data sharing requirements. PLOS does not

permit references to inaccessible data. We require that authors provide all relevant data within the paper, Supporting Information files, or in an acceptable, public

repository. Please add a citation to support this phrase or upload the data that corresponds with these findings to a stable repository (such as Figshare or Dryad) and provide and URLs, DOIs, or accession numbers that may be used to access these data. Or, if the data are not a core part of the research being presented in your study, we ask that you remove the phrase that refers to these data.

In the previous submission, we had two places used “data not shown”. One place states “Btz treatment did not alter the percentage of B220+ B cells, CD3+ T cells and CD11b+ monocytes in the same samples (data not shown)”. Another place states “targeting a single E3 ligase may not be efficient, especially since we have found levels of multiple E3 ligases elevated in fracture callus and Itch deficient mice have similar fracture repair as WT littermates (data not shown)”. In the revised manuscript, we have included these data in Figure 5B and S1 Fig.

6. Your ethics statement should only appear in the Methods section of your manuscript. If your ethics statement is written in any section besides the Methods, please

delete it from any other section.

In the revised manuscript, ethics statement is only in the Methods Section.

7. Please include a copy of Table 1 which you refer to in your text on page 7.

We have included S1 Table in the revised manuscript. (page 8, line 234)

Additional Editor Comments:

This study needs to be considerably amended for English. A native speaker expert in Scientific English must revise it in deep.

We have extensive edited English in the revised manuscript.

Results section should must be focused ONLY to the obtained results: is this the first manuscript they write? Therefore please move all comments to the Discussion Section.

We have extensively revised result section and depleted or moved the comments to the Discussion Section.

Authors must analyze several other osteogenic markers, such as BSP, OPN, OC, both for qRT-PCR and WB.

We have examined the expression levels of additional osteogenic markers, bone sialoprotein, osteopontin and osteocalcin by qPCR, and osteocalcin protein levels by Western blot analysis as reviewer suggested. We have included these new data in Figure 2D and 2E and in text. (page 10, line 290-294)

Better images related to colony formation in figure 4B MUST be added: they are not available.

We decided to remove colony formation image from the figure 4B because we have eliminated Nestin-GFP mouse colony in University of Rochester Medical Center due to limited budget and Covid 19. Since more recent publication have further characterized Nestin-GFP cells as skeletal or perivascular mesenchymal progenitor cells [1-4] we believe that removal these data do not affect the overall conclusion of our current study. 

Micro-Computed Tomography (μCT) and Radiological Analysis must also be added to increase the interest to this paper.

We have included μCT data for this study in Figure 2C and in text. (page 10, line 290)

We do not use X-ray in fracture study because X-ray is not an ideal and quantitative method to monitor the fracture healing process. The Center for Musculoskeletal Research, University of Rochester Medical Center, has an established SOP to evaluate the fracture healing process, including histology, micro-CT and bio-mechanical testing [5-7]. 

Reviewers' comments:

Reviewer's Responses to Questions

Comments to the Author

1. Is the manuscript technically sound, and do the data support the conclusions?

The manuscript must describe a technically sound piece of scientific research with data that supports the conclusions. Experiments must have been conducted

rigorously, with appropriate controls, replication, and sample sizes. The conclusions must be drawn appropriately based on the data presented.

Reviewer #1: Partly

2. Has the statistical analysis been performed appropriately and rigorously?

Reviewer #1: Yes

3. Have the authors made all data underlying the findings in their manuscript fully available?

The PLOS Data policy requires authors to make all data underlying the findings described in their manuscript fully available without restriction, with rare exception

(please refer to the Data Availability Statement in the manuscript PDF file). The data should be provided as part of the manuscript or its supporting information, or

deposited to a public repository. For example, in addition to summary statistics, the data points behind means, medians and variance measures should be available. If

there are restrictions on publicly sharing data—e.g. participant privacy or use of data from a third party—those must be specified.

Reviewer #1: Yes

4. Is the manuscript presented in an intelligible fashion and written in standard English?

PLOS ONE does not copyedit accepted manuscripts, so the language in submitted articles must be clear, correct, and unambiguous. Any typographical or grammatical

errors should be corrected at revision, so please note any specific errors here.

Reviewer #1: No

5. Review Comments to the Author

Please use the space provided to explain your answers to the questions above. You may also include additional comments for the author, including concerns about dual

publication, research ethics, or publication ethics. (Please upload your review as an attachment if it exceeds 20,000 characters)

Reviewer #1: In this study Authoirs examined the expression pattern of the Nedd4 sub-class of E3s and Ub proteins in fracture callus and the effect of Bortezomib on

fracture repair on a tibial fracture model in Nestin-GFP reporter mice.

English should be revised. In particular, in paragraph: “Tibial fracture model, Bortezomib treatment, and callus preparations.” Authors used different time.

We have extensive edited English in the revised manuscript.

More references are needed in the first part of introduction.

We have added more references in the first part of introduction. 

“Bortezomib and Carfilzomib are 2 FDA-approved drugs”: please change 2 with two.

Results section should be revised. They should be reduced to results obtained. All the comments should be moved to the discussion section.

 We have changed “2” to “two” on page 4. We 

Authors should analyze other osteogenic markers, whose expression is related to that of Runx2, such as BSP, OPN, OC, both for qRT-PCR and WB.

We have examined the expression levels of additional osteogenic markers, bone sialoprotein, osteopontin and osteocalcin by qPCR, and osteocalcin protein levels by Western blot analysis as reviewer suggested. We have included these new data in Figure 2D and 2E and in text. (page 10, line 290-294)

Better images of colony formation in figure 4B should be provided.

We decided to remove colony formation image from the figure 4B because we have eliminated Nestin-GFP mouse colony in University of Rochester Medical Center due to limited budget and Covid 19. Since more recent publication have further characterized Nestin-GFP cells as skeletal or perivascular mesenchymal progenitor cells [1-4] we believe that removal these data do not affect the overall conclusion of our current study. 

Micro-Computed Tomography (μCT) and Radiological Analysis would add interest to the paper.

We have included μCT data for this study (Figure 2C, text page 10, line 290). 

We do not use X-ray in fracture study because X-ray is not an ideal and quantitative method to monitor the fracture healing process. The Center for Musculoskeletal Research, University of Rochester Medical Center, has an established SOP to evaluate the fracture healing process, including histology, micro-CT and bio-mechanical testing [5-7]. 

6. PLOS authors have the option to publish the peer review history of their article. If published, this will include your full peer review and any attached files.

Do you want your identity to be public for this peer review? For information about this choice, including consent withdrawal, please see our Privacy Policy.

Reviewer #1: No

While revising your submission, please upload your figure files to the Preflight Analysis and Conversion Engine (PACE) digital diagnostic tool, https://pacev2.apexcovantage.com/. PACE helps ensure that figures meet PLOS requirements. To use PACE, you must first register as a user. Registration is free.

Then, login and navigate to the UPLOAD tab, where you will find detailed instructions on how to use the tool. If you encounter any issues or have any questions when using PACE, please email PLOS at figures@plos.org. Please note that Supporting Information files do not need this step.

References 

1. Wang S, Su TT, Tong H, Shi W, Ma F, Quan Z. CircPVT1 promotes gallbladder cancer growth by sponging miR-339-3p and regulates MCL-1 expression. Cell Death Discov. 2021;7(1):191. Epub 2021/07/28. doi: 10.1038/s41420-021-00577-y. PubMed PMID: 34312371; PubMed Central PMCID: PMCPMC8313687.

2. Tournaire G, Stegen S, Giacomini G, Stockmans I, Moermans K, Carmeliet G, et al. Nestin-GFP transgene labels skeletal progenitors in the periosteum. Bone. 2020;133:115259. Epub 2020/02/10. doi: 10.1016/j.bone.2020.115259. PubMed PMID: 32036051.

3. Nakahara F, Borger DK, Wei Q, Pinho S, Maryanovich M, Zahalka AH, et al. Engineering a haematopoietic stem cell niche by revitalizing mesenchymal stromal cells. Nat Cell Biol. 2019;21(5):560-7. Epub 2019/04/17. doi: 10.1038/s41556-019-0308-3. PubMed PMID: 30988422; PubMed Central PMCID: PMCPMC6499646.

4. Mendez-Ferrer S, Michurina TV, Ferraro F, Mazloom AR, Macarthur BD, Lira SA, et al. Mesenchymal and haematopoietic stem cells form a unique bone marrow niche. Nature. 2010;466(7308):829-34. Epub 2010/08/13. doi: 10.1038/nature09262. PubMed PMID: 20703299; PubMed Central PMCID: PMCPMC3146551.

5. Shares BH, Smith CO, Sheu TJ, Sautchuk R, Jr., Schilling K, Shum LC, et al. Inhibition of the mitochondrial permeability transition improves bone fracture repair. Bone. 2020;137:115391. Epub 2020/05/04. doi: 10.1016/j.bone.2020.115391. PubMed PMID: 32360587; PubMed Central PMCID: PMCPMC7354230.

6. Wang H, Zhang H, Srinivasan V, Tao J, Sun W, Lin X, et al. Targeting Bortezomib to Bone Increases Its Bone Anabolic Activity and Reduces Systemic Adverse Effects in Mice. J Bone Miner Res. 2020;35(2):343-56. Epub 2019/10/15. doi: 10.1002/jbmr.3889. PubMed PMID: 31610066.

7. Wang C, Inzana JA, Mirando AJ, Ren Y, Liu Z, Shen J, et al. NOTCH signaling in skeletal progenitors is critical for fracture repair. J Clin Invest. 2016;126(4):1471-81. Epub 2016/03/08. doi: 10.1172/JCI80672. PubMed PMID: 26950423; PubMed Central PMCID: PMCPMC4811137.

---

## [Decision Letter · Decision Letter 1]

28 Jan 2022

Proteasome Inhibition-Enhanced Fracture Repair is Associated with Increased Mesenchymal Progenitor Cells in Mice

PONE-D-21-32074R1

Dear Dr. Liu,

We’re pleased to inform you that your manuscript has been judged scientifically suitable for publication and will be formally accepted for publication once it meets all outstanding technical requirements.

Kind regards,

Gianpaolo Papaccio, M.D., Ph.D.

Academic Editor

PLOS ONE

Additional Editor Comments (optional):

All previous comments have been addressed

Reviewers' comments:

Reviewer's Responses to Questions

**Comments to the Author**

1. If the authors have adequately addressed your comments raised in a previous round of review and you feel that this manuscript is now acceptable for publication, you may indicate that here to bypass the “Comments to the Author” section, enter your conflict of interest statement in the “Confidential to Editor” section, and submit your "Accept" recommendation.

Reviewer #1: All comments have been addressed

2. Is the manuscript technically sound, and do the data support the conclusions?

Reviewer #1: (No Response)

3. Has the statistical analysis been performed appropriately and rigorously? 

Reviewer #1: (No Response)

4. Have the authors made all data underlying the findings in their manuscript fully available?

Reviewer #1: (No Response)

5. Is the manuscript presented in an intelligible fashion and written in standard English?

Reviewer #1: (No Response)

6. Review Comments to the Author

Reviewer #1: (No Response)

7. PLOS authors have the option to publish the peer review history of their article (what does this mean?). If published, this will include your full peer review and any attached files.

Reviewer #1: No

---

## [Editor Report · Acceptance letter]

14 Feb 2022

PONE-D-21-32074R1 

Proteasome Inhibition-Enhanced Fracture Repair is Associated with Increased Mesenchymal Progenitor Cells in Mice 

Dear Dr. Liu:

I'm pleased to inform you that your manuscript has been deemed suitable for publication in PLOS ONE. Congratulations! Your manuscript is now with our production department. 

Kind regards, 

on behalf of

Prof. Gianpaolo Papaccio 

Academic Editor

PLOS ONE